# Towards Mixed Optimization for
# Reinforcement Learning with Program Synthesis

## Abstract

Deep reinforcement learning has led to many recent breakthroughs, but the learnt policies are often based on black-box neural networks, which makes them difficult to interpret and to impose desired specification constraints during learning. We present an iterative framework, MORL, for improving the learned policies using program synthesis. Concretely, we propose to use synthesis techniques to obtain a symbolic representation of the learned policy, which can then be debugged manually or automatically using program repair. After the repair step, we use behavior cloning to obtain the policy corresponding to the repaired program, which is then further improved using gradient descent. This process continues until the learned policy satisfies desired constraints. We instantiate MORL for the simple CartPole problem and show that the programmatic representation allows for high-level modifications that in turn lead to improved learning of the policies.

## 1. Introduction

There have been many recent successes in using deep reinforcement learning (DRL) to solve challenging problems such as learning to play Go and Atari games (Silver et al., 2016; 2017; Mnih et al., 2015). While the effectiveness of these reinforcement learning methods in these domains has been impressive, some shortcomings for these learned policies based on black-box deep neural networks are that they are difficult to interpret and that it is challenging to impose and validate certain desirable policy specifications, such as worst-case guarantees or safety constraints. This make it difficult to debug and improve these policies and therefore hinders their use for safety-critical domains.

There has been some recent work on using program synthesis techniques to interpret learned policies using higher-level programs (Verma et al., 2018) and decision trees (Bastani et al., 2018). The key idea in PIRL (Verma et al., 2018) is to first train a DRL policy using standard methods and then use an imitation learning-like approach to search for a program in a domain-specific language (DSL) that conforms to the behavior traces sampled from the policy. Similarly, VIPER (Bastani et al., 2018) uses imitation learning (a modified form of the DAGGER algorithm (Ross et al., 2011)) to extract a decision tree corresponding to the learned policy. The main goal of these works is to extract a symbolic high-level representation of the policy as a DSL program or a decision tree that is more interpretable and also amenable for program verification techniques.

We build upon these recent advances to propose an iterative framework for learning interpretable and safe policies. The main steps in the workflow of our framework are as follows. We first start with an initial random policy $\pi_0$. We then use program synthesis techniques similar to PIRL and VIPER to learn a symbolic representation of the learned policy as a program $P_0$. After obtaining a programmatic representation of the policy, we then perform program repair (Weimer et al., 2009; Jobstmann et al., 2005) to obtain a repaired program $P_0'$ that satisfies some set of constraints. Note that the program repair step can be performed either automatically using a safety specification constraint or it can be performed manually by a human expert that modifies $P_0$ to remove undesirable behaviors (or add desired behaviors). We then use behavioral cloning (Bratko et al., 1995) to obtain the corresponding improved policy $\pi_0'$, which is then further improved using standard gradient descent to obtain $\pi_1$. This process of improving policies from $\pi_t \rightarrow P_t \rightarrow P_t' \rightarrow \pi_t' \rightarrow \pi_{t+1}$ is repeated iteratively until achieving desirable performance and safety guarantees. We name this iterative procedure a mixed optimization scheme for reinforcement learning, or MORL.

As a first step towards a full realization of MORL, we present a simple instantiation of our framework for the Cart-Pole (Barto et al., 1983) problem, where we demonstrate the efficacy of our approach to learn near-optimal policies, while enabling the user to better interpret the learned policy.

[1]Anonymous Institution, Anonymous City, Anonymous Region, Anonymous Country. Correspondence to: Anonymous Author <anon.email@domain.com>.

Preliminary work. Under review by the International Conference on Machine Learning (ICML). Do not distribute.

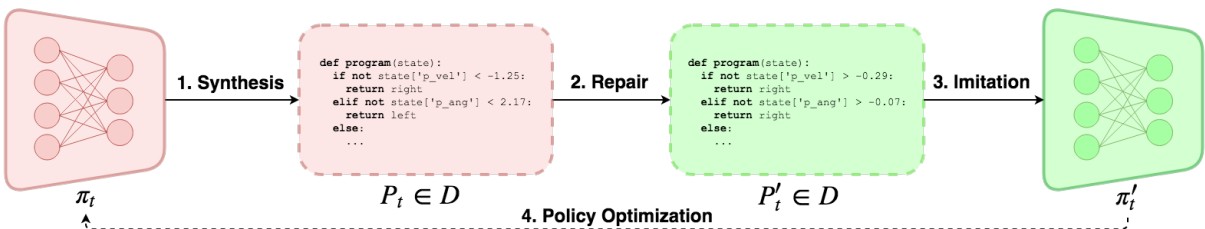

*Figure 1.* An overview of the proposed method. We decompose policy learning into alternating between policy optimization and program repair. Starting from a black-box policy $\pi_t$, we consider the following steps (1) **Synthesis**, which generates a program $P_t$ corresponding to the policy $\pi_t$. The program is sampled from an underlying Domain Specific Language (DSL) $\mathcal{D}$ (2) **Repair**, which corresponds to *debugging* the program ,which allows us to impose high-level constraints on the learned program. (3) **Imitation** corresponds to distilling the program back into a reactive representation. (4) **Policy Optimization** in this case corresponds to gradient-based policy optimization.

In addition, we argue that the scheme has a natural interpretation and can be readily extended to capture more notions of policy improvement and discuss the potential benefits and obstacles of using such an approach.

This paper makes the following key contributions:

- We propose a simple framework for iterative policy refinement by performing repair at the level of programmatic representation of learned policies.
- We instantiate the framework for the CartPole problem and show the effectiveness of performing modifications in the symbolic representation of the learned policies.

## 2. Mixed Optimization for Reinforcement Learning

Our goal is to improve policy learning by decomposing the usual gradient-based optimization scheme into an iterative two-stage algorithm. In this context, we can view improvement as either making the policies (1) *safe* – to ensure performance under safety, (2) *interpretable* – allowing some level of introspection into the policy's decisions, (3) *sample efficient*, or (4) *alignment* with priors. While there are other notions of improvement, for the remainder of the paper, we focus on sample efficiency as a notion of policy improvement. We include a discussion of the other approaches as they apply to our framework.

### 2.1. Problem Definition

Consider the typical Markov decision process (MDP) setup $(\mathcal{S}, \mathcal{A}, \mathcal{R}, \mathcal{T}, \rho_0, \gamma)$, with a state space $\mathcal{S}$, an action space $\mathcal{A}$, a reward function $\mathcal{R}$, the transition dynamics of the environment $\mathcal{T}$, the initial starting state distribution $\rho_0$, and the discount factor $\gamma$. The goal will be to find a policy, or function $\pi : \mathcal{S} \to \mathcal{A}$, that achieves the maximum expected reward. Normally, the reward design and specification for a task **T** corresponds to defining the reward function $\mathcal{R}(s, a)$,

such that an optimal policy $\pi^*$ *solves* the task.

An alternative view of solving the task could be defined as having access to an oracle policy $\pi$ or a fixed number of trajectories from it. In this setting, our goal is learning a policy by *imitation learning*, which would also equivalently *solve* the task. In this work, we focus on improving policy learning using imitation learning (Abbeel & Ng, 2004; Ho & Ermon, 2016), though the framework is more general and extends well to reinforcement learning.

We consider a symbolic representation $\mathcal{D}$ (such as a DSL) that is expressive enough to represent different policies. The synthesis problem can then be defined as learning a program $P \in \mathcal{D}$ such that $\forall s \in \mathcal{S} : \pi(s) \approx P(s)$, i.e. the learned program $P$ produces approximately the same output actions as the actions produced by the policy $\pi$ for all (or a sampled set of) input states $S$.

### 2.2. Model

In MORL we maintain two representations of a policy:

- a **reactive, black-box policy** where we represent the policy as a differentiable function, such as a neural network, allowing us to use gradient-based optimization methods like TRPO (Schulman et al., 2015) or PPO (Schulman et al., 2017).
- a **symbolic program**, which represents the policy as an interpretable program. The symbolic program representations are amenable for analysis and transformations using automated program verification and repair techniques, or human inspection.

With these intermediate representations, we are able to alternate between the following; the first allows us to finetune poicies in a function space, while the second allows us to impose hard constraints or incorporate human debugging. The general procedure (shown in Fig 1) consists of four key steps as detailed below.

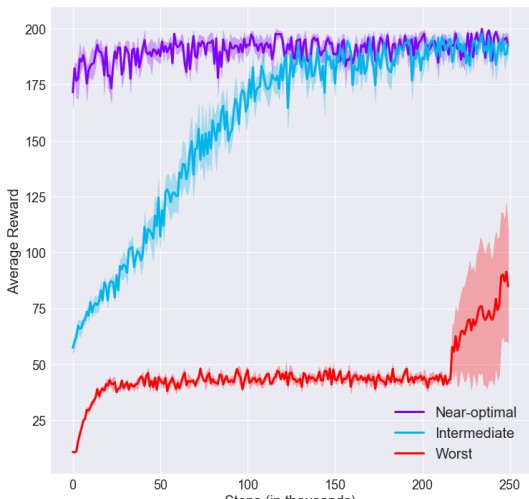
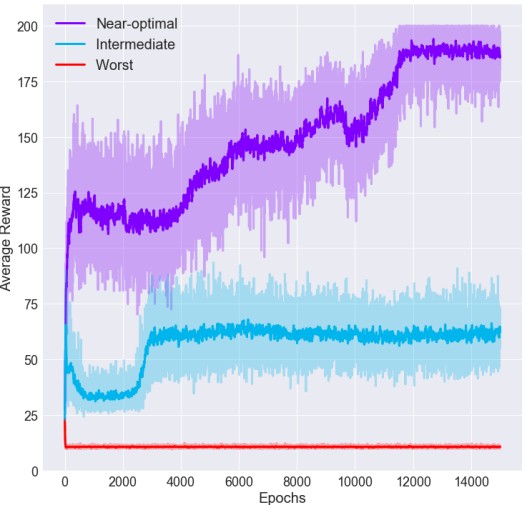

*Figure 2.* Evaluating the usefulness of maintaining differentiable, and symbolic representations of the policy. Each plot corresponds to finetuning a policy cloned from a program (in this case decision trees) with TRPO (averaged over 5 runs). In this case, `Near-Optimal` is obtained by *manual debugging* of the `Intermediate` policy, which is obtained from `Worst` policy.

**Synthesis**: Given a task **T**, we consider a Domain Specific Language $\mathcal{D}$, such that there exists some program $P \in \mathcal{D}$ that is a sufficient representation of the task. In the first step of MORL, we seek to synthesize such a program that is equivalent to the corresponding policy $\pi$. A programmatic representation of the policy allows us to leverage approaches such as program repair and verification to provide guarantees for the underlying policy. For this step, and in the scope of this paper, we assume that we can utilize existing program synthesis methods such as VIPER or PIRL, so we do not attempt to perform this step explicitly. We focus on the following steps in the MORL scheme.

**Repair**: In this step, we modify the synthesized program accordingly to satisfy constraints imposed either on $\mathcal{D}$, or on the synthesized program $P$. This step allows us to meaningfully *debug* the policy, either through human-in-the-loop verification for interpretability, or through automated program repair techniques that involve defining Constraint Satisfaction Problems (CSP) typically solved using SAT/SMT solvers (Singh et al., 2013). For the scope of this paper, we mimic the repair process by manually modifying the initial program to obtain three programs that achieve three different levels of success at the task of interest.

**Imitation**: Following the program synthesis and repair steps, we then distill the program back into a reactive policy using imitation learning. Given that we have access to an oracle $P'_t$, we find that we reliably imitate the program (Ross et al., 2011). Note that it is possible to stop the optimization here. Indeed, we observe that a user may end the optimiza-

*Figure 3.* An important step in the algorithm is alternating between symbolic and policy representations. Here we plot the convergence rate of *randomly initialized* policies to the program behavior. In this work, we used simple behavioral cloning to retrain the policies. We note that more sample efficient algorithms would be able to emulate the behavior from the program more quickly.

tion procedure here of MORL if a certain performance or safety bound has been reached, and may skip the last step.

**Policy Optimization** Finally, we may finetune the policy using gradient descent. We posit that by optimizing in both program space and over the space of policies in a differentiable space, we are able to better escape local minima while still maintaining an underlying intuition for how the policy is performing from the inspection of the program.

## 3. Experiments

We evaluate our framework on the CartPole-v0 problem in the OpenAI Gym environment for discrete control (Brockman et al., 2016). We present a first simple instantiation of the framework to showcase its usefulness compared to direct reinforcement learning. In our preliminary evaluation, we evaluate the following research questions:

- Does program repair lead to faster convergence?

- Does programmatic representation help humans provide better repair insights?

To this effect, we train an initial policy $\pi_0$ (`Worst`) that performs poorly, and then extract the corresponding symbolic representation $P_0$. For the symbolic representation, we chose VIPER's (Bastani et al., 2018) decision tree representation of the policy. We then modify the symbolic program to get a new program $P'_0$, which performs better than the original program by repairing certain values in the decision tree. This is followed by behavioural cloning to obtain $\pi'_0$

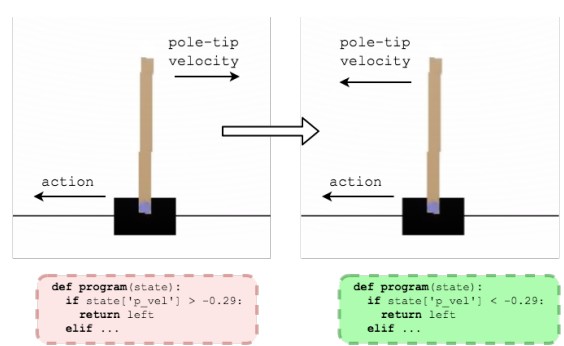

*Figure 4.* Debugging `Worst` (red) to `Intermediate` (green). In one step of debugging the policy, we fix the policy to make the cart shift in the same direction as the pole.

(corresponding to $P_0'$), which is optimized to obtain $\pi_1$.

To simulate the iterative optimization of the framework, we perform two different modifications of the program repair step to obtain $P_0^1$ (`Intermediate`) and $P_0^2$ (`Near-optimal`) that have different characteristics in terms of repair improvements. For example, the modification to obtain program $P_0^1$ from $P_0$ is shown in Fig 4, where we manually provide the insight of making the cart shift in the same direction as the pole.

In our experiments, we first find the average performance of each of the levels of policies across 25 runs. The `Worst` policy gets an average reward of 9.28, the `Intermediate` policy gets an average reward of 104.0, and the `Near-optimal` policy gets an average reward of 200.0. When we attempt to distill the programs to continuous policies $\pi$, we find that each of the resulting levels of policies get 10.64, 66, and 185, respectively, as shown in Figure 3 after 15000 epochs. Lastly, when we take the resulting distilled policies and then finetune these with TRPO, we find that the resulting average rewards are 38.65, 79.03, and and 176.8 after 25 episodes of training with 10 trajectories of length 200. In Figure 2, we run TRPO for a total of 250 episodes to see the limiting behavior.

From our results, we validate our hypothesis that under bad initialization (`Worst`), TRPO takes an order of magnitude longer to converge to near-optimal policy, when compared to policies initialized after program repair. We believe that providing high-level insights programmatically can help policies discover better or safer behaviors.

## 4. Related Work

Our framework is inspired from the recent works of PIRL (Verma et al., 2018) and VIPER (Bastani et al., 2018) in using program synthesis techniques to learn symbolic interpretable representations of the learnt policies, and then using program verification techniques to verify certain prop-

erties of the program.

PIRL first trains a DRL policy for a domain and then uses an imitation learning like approach to generate specifications (input-output behaviors) for the synthesis problem. It then uses a Bayesian optimization technique to search for programs in a DSL that conforms to the specification. It iteratively builds up new behaviors by executing the initial policy as an oracle to obtain outputs for inputs that were not originally sampled but are observed in executing the learnt programs. It maintains a family of programs consistent with the specification and chooses the one as output that achieves the maximum reward on the task.

VIPER uses a modified form of the DAGGER initiation learning algorithm to extract a decision tree corresponding to the learnt policy. It then uses program verification techniques to validate correctness, stability, and robustness properties of the extracted programs (represented as decision trees).

While previous approaches stop at learning a verifiable symbolic representation of policies, our framework aims at iterative improvement of the policies. In particular, if the extracted symbolic program does not satisfy certain desirable verification constraints, unlike previous approaches, our framework allows for repairing the programs in the symbolic space and then distilling the programs to policies for further optimization.

## 5. Discussion and Future Work

We presented a preliminary instantiation of the MORL framework showing the benefits of learning a symbolic representation of the policy. Namely, that by optimizing the policy by iterating between two representations, we were able to converge faster to near-optimal performance starting with a poor initialization.

There are a number of assumptions we make in this paper in order to instantiate our framework. While the MORL framework is general enough to encapsulate many different approaches of synthesis, repair, and imitation, we only consider the simplest forms of these. For instance, we hand-design the candidate repaired programs, and use a simple supervised approach for imitation learning. Each of these aspects could be significantly scaled up to be used for larger programs and for more complicated tasks. While CartPole was a simple sandbox for which we could test symbolic programs, for more complicated tasks, automated program repair and verification techniques would be more efficient.

Reward design and safety (Hadfield-Menell et al., 2017) is another exciting research direction. Note that we can instead use the reward function $\mathcal{R}$ as the program representation for MORL; this would instead provide a procedure for more interpretable or verifiable inverse reinforcement learning.

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
