# OpenReview forum: "Towards Mixed Optimization forReinforcement Learning with Program Synthesis"
_ICML.cc/2018/Workshop/NAMPI — NAMPI 2018_

### Review · AnonReviewer2 · 2018-06-21
**Ambitious, but suitable**

**Rating:** 7
**Confidence:** 4

**Review:**

This paper builds on the policy-distillation-as-program-synthesis approaches which have appeared in the literature in the last year to propose an innovative and interpretable training loop. A deep RL policy (presumably pre-trained against an environment objective) is distilled into an interpretable program (e.g. using VIPER). Said program is either repaired by automated methods, or through human intervention (by virtue of it being human readable). The neural policy is then adjusted through imitation learning to imitate the programmatic policy (why not just do policy distillation here, though?), and then fine-tuned against the original objective. It's a nice idea, for which there is a very toy and idealised experiment, which is nonetheless sufficient to prove the concept. I think while at a high level, this makes a lot of sense, this proposal lives or dies by the scalability of policy-distillation-as-program-synthesis approaches to non-trivial Deep RL policies, which remains to be seen. That said, the exploratory nature of this paper and its focus on exploiting the interpretability of programs make it suitable for NAMPI.

---

### Review · AnonReviewer3 · 2018-06-25
**Good paper introducing an important direction in program synthesis for ML**

**Rating:** 7
**Confidence:** 4

**Review:**

The authors present an iterative framework (MORL) for mixing neural reinforcement learning (RL) with program synthesis. Namely: first, one synthesises a neural policy, then one makes a program to explain it, finally, one repairs the program to get better behaviour, and then the cycle continues. The authors show that this approach improves policies on the simple cartpole environment. This is an important direction as interpretability is a great challenge in deep learning and RL and it's a great contribution to show that it can be made to work end-to-end even in the simple setting of cartpole. It is unclear if the program synthesis and repair steps can be scaled up to more complex polices at all, but that remains to be seen in future work.

---

### Decision · ~NAMPI_Admin1 · 2018-06-28
**Paper7 Final Decision**

Accept